# Detection of Crimean–Congo Haemorrhagic Fever Virus from Livestock Ticks in Northern, Central and Southern Senegal in 2021

**DOI:** 10.3390/tropicalmed8060317

**Published:** 2023-06-12

**Authors:** Aminata Badji, Mignane Ndiaye, Alioune Gaye, Idrissa Dieng, El Hadji Ndiaye, Anna S. Dolgova, Moufid Mhamadi, Babacar Diouf, Ibrahima Dia, Vladimir G. Dedkov, Oumar Faye, Mawlouth Diallo

**Affiliations:** 1Pôle de Zoologie Médicale, Institut Pasteur de Dakar, Dakar 220, Senegal; aminassbadji@gmail.com (A.B.); alioune.gaye@pasteur.sn (A.G.); elhadji.ndiaye@pasteur.sn (E.H.N.); babacar.diouf2@pasteur.sn (B.D.); ibrahima.dia@pasteur.sn (I.D.); 2Pôle de Virologie, Institut Pasteur de Dakar, Dakar 220, Senegal; mignane.ndiaye@pasteur.sn (M.N.); idrissa.dieng@pasteur.sn (I.D.); moufid.mhamadi@pasteur.sn (M.M.); oumar.faye@pasteur.sn (O.F.); 3St. Petersburg Pasteur Institute, Federal Service for Consumer Rights Protection and Human Well-Being Surveillance, St. Petersburg 190013, Russia; annadolgova@inbox.ru (A.S.D.); vgdedkov@yandex.ru (V.G.D.)

**Keywords:** CCHFV, livestock, ticks, Senegal

## Abstract

Crimean–Congo haemorrhagic fever virus (CCHFV) occurs sporadically in Senegal, with a few human cases each year. This active circulation of CCHFV motivated this study which investigated different localities of Senegal to determine the diversity of tick species, tick infestation rates in livestock and livestock infections with CCHFV. The samples were collected in July 2021 from cattle, sheep and goats in different locations in Senegal. Tick samples were identified and pooled by species and sex for CCHFV detection via RT-PCR. A total of 6135 ticks belonging to 11 species and 4 genera were collected. The genus *Hyalomma* was the most abundant (54%), followed by *Amblyomma* (36.54%), *Rhipicephalus* (8.67%) and *Boophilus* (0.75%). The prevalence of tick infestation was 92%, 55% and 13% in cattle, sheep and goats, respectively. Crimean–Congo haemorrhagic fever virus (CCHFV) was detected in 54/1956 of the tested pools. The infection rate was higher in ticks collected from sheep (0.42/1000 infected ticks) than those from cattle (0.13/1000), while all ticks collected from goats were negative. This study confirmed the active circulation of CCHFV in ticks in Senegal and highlights their role in the maintenance of CCHFV. It is imperative to take effective measures to control tick infestation in livestock to prevent future CCHFV infections in humans.

## 1. Introduction

Ticks are the first and second most important vectors of animal and human diseases, respectively [1]. In the context of global and environmental changes, many tropical and subtropical countries are facing major public health crises related to the emergence or re-emergence of tick-borne diseases. These phenomena are explained by the expansion into new geographical areas suitable for the development of tick vectors and/or reservoir hosts [2]. Factors contributing to the emergence or re-emergence of zoonoses include adaptation to the host’s immune system, rapid spread through livestock and migratory bird movements, and widespread invasion of natural habitats by humans through their economic activities (agriculture, mining, fruit or honey collection, road construction, and extensive livestock farming) [3]. In response to this situation, several studies have been conducted to determine the diversity of ticks and the viruses they transmit, with a focus on the Crimean–Congo haemorrhagic fever virus (CCHFV) [4].

Crimean–Congo haemorrhagic fever (CCHF) is a disease caused by a tick-borne virus of the order Bunyavirales, family *Nairoviridae* and genus *Orthonairovirus* [5]. It has the widest geographic distribution of all tick-borne viral diseases, being found in more than 30 countries in Africa, Asia, Eastern Europe and the Middle East [6]. Crimean–Congo haemorrhagic fever virus circulates mainly in rural areas. As with all vector-borne diseases, the presence and persistence of zoonotic outbreaks of CCHF depend on the biological and ecological relationships between three types of organisms: the virus, the tick and the vertebrate hosts [7]. Although CCHFV has been isolated from several species of hard ticks (Ixodidae), the primary vector group appears to be ticks of the genus *Hyalomma* [8].

Tick bites remain the primary mode of transmission of CCHFV to humans and animals. However, human infections can also occur via direct contact with infected blood, fluids or other tissues of animals or patients during the acute phase of the disease. Animals are asymptomatic reservoirs of the virus. However, infection in humans can cause a non-specific febrile illness characterised by fever, myalgia, diarrhoea, nausea and vomiting after a short incubation period (1–5 days). In some cases, symptoms may progress to severe haemorrhagic disease [9]. 

In Senegal, evidence of CCHFV circulation has been reported since 1960 [10]. Between 2003 and 2020, nine confirmed cases and one death were recorded in different regions (Matam, Thiès, Saint-Louis, Kaolack, Fatick, and Tambacounda) [11]. More recently, eight confirmed cases and two deaths (between 2021 and 2022) were reported (unpublished data). Epidemiological and environmental data indicate that the northern regions of Senegal are at the highest risk of CCHF outbreaks [12]. This may be due to their ecological parameters and proximity to Mauritania where the CCHF epidemic is recurrent [13,14]. Indeed, the most recent strains/isolates were detected in Northern Senegal in the Matam region. Additionally, genetic characterisation showed that some of these strains belong to the same group as a strain isolated in Mauritania in 1984. 

The present study was initiated and conducted in this national context of active CCHFV circulation in several localities in Senegal. The main objective was to investigate emerging and re-emerging tick-borne viruses circulating in Senegal. More specifically, the aim was to create an inventory of the diversity of ticks infesting livestock (cattle, goats and sheep) but also to estimate the infection rate of CCHFV in ticks in several localities in the north, centre and south of Senegal where CCHF cases have been reported.

## 2. Materials and Methods

### 2.1. Collection Sites

The study was conducted in different geographical regions (Table 1) from the north to the south of the country (Figure 1). The climate is subtropical, with a long dry season (approximately mid-October to mid-June in the north and early November to mid-May in the south) and a rainy season (approximately, late June to early October in the north, and late May to late October in the south). The north of the country (Nabadji, Haere Lao, Tatki, Tessekere, Kassack and Saint-Louis) is considered a semi-desert area with an annual rainfall of about 300 mm [15]. In these areas, the main economic activity of the populations is nomadic pastoralism with livestock including cattle, sheep and goats. The long dry seasons cause people and livestock to move from the north to the south of the country, where water and pasture are abundant after the rainy season. The other sites are located in the centre of the country (Bandia), with wooded savannah, and in the south (Kolda), a forested area characterised by an annual rainfall of up to 1000 mm. These collection sites were chosen because of the high prevalence of ticks on livestock and the isolation of CCHFV from ticks collected in these areas in the past.

### 2.2. Tick Sampling

Sampling was carried out from 1 to 14 July 2021. We visited 24 herds with a total of 720 animals, including 240 cattle, 240 sheep and 240 goats kept under semi-extensive conditions. The animals were kept in cattle sheds and sheepfolds at night, and during the day they were turned out onto pastures around the villages and drink from the watering point. An average of 30 animals were randomly selected from each farm during each visit. Tick collections were carried out on domestic animals. All ticks, irrespective of their stage of development (adults, nymphs, or larvae) were collected via extirpation using hard forceps from the entire body of the animal (ear, head, neck, back, fanon, abdomen, anogenital region, tail and feet). The tick was pulled firmly by the hypostome to avoid damage that could make species identification difficult. Once collected, the ticks were placed in 2 mL microfuge tubes labelled with the date and location of collection, ID number, sex of animal and site of tick attachment. All tubes were then stored in liquid nitrogen at −180 °C until treatment.

### 2.3. Identification and Processing of Tick Samples

In the laboratory, the ticks were washed with sterile water and then identified on a chill table under a binocular stereo-zoom microscope using appropriate morphological characteristics [16,17]. After identification, the ticks were grouped into monospecific pools of 1 to 28 individuals according to sex, date, collection site and feeding status. Each pool was triturated in 0.5 or 2 mL of L15 medium (10% foetal bovine serum, 100 U/mL penicillin, 100 μg/mL streptomycin, and 1 μL/mL amphotericin B) under BSL-3 laboratory conditions using a sterile bead homogeniser (Tissue Lyser II—QIAGEN, Hilden, Germany), depending on size. The tick homogenates were centrifuged at 2500 rpm for 5 min at 4 °C and the supernatants were stored at −80 °C until use.

### 2.4. Detection of CCHFV by RT-PCR

RNA was extracted from the tick supernatant using QIAamp RNA Viral kit (Qiagen GmbH, Heiden, Germany) according to the manufacturer’s recommendations. The RNA was eluted in 60 μL of AVE buffer and stored at −80 °C until use. Detection of CCHFV was performed using the AmpliSens CCHFV-FRT PCR kit (Amplisens, Bratislava 47, Slovak Republic) according to the manufacturer’s recommendations. A total of 10 μL of RNA was added to a 15 μL reaction mixture consisting of 12.5 μL of the buffer, 4 μL of nuclease-free water, 1 μL of each primer, 0.5 μL of the probe and 1 μL of the enzyme. The qRT- PCR was performed on CFX96 (Biorad, Biorad Laboratories, Marnes-La-Coquette, France). The cycling conditions were 50 °C for 30 min and 95 °C for 15 min, followed by 5 cycles of 95 °C for 10 s, 54 °C for 30 s and 72 °C for 15 s and finally 45 cycles of 95 °C for 10 s, 50 °C for 30 s (where signal acquisition was performed) and 72 °C for 15 s. The signal of the CCHFV cDNA amplification product was detected in the channel for the JOE fluorophore.

### 2.5. Data Analysis

The number of ticks collected from animals was recorded by collection date, sampling location and animal species in a 2016 Excel spreadsheet. The relative abundance for a sampling location was calculated by dividing the number of ticks collected at that site by the total number of ticks sampled. For each tick species, the relative abundance was calculated as the ratio of the total number of ticks collected from a vertebrate host of that species to the total number of species collected from all hosts inspected at a given site. The tick infestation prevalence (IP) was the ratio of the total number of infested animals to the total number of animals examined [18]. The X^2^ test was used to compare IP and relative abundance values. The biodiversity of the species collected at the different sites was estimated using the following indices.

One index is species richness, which is the total number of species in a sample. The other is the Shannon–Weaver index [19], which quantifies the number of species collected and their relative abundance, and measures the heterogeneity of an environment or a study period. It is calculated using the following formula:$$\;H^{\prime }=-\overset{S}{\underset{i=1}{\sum }}\;pi\;\ln pi$$
where H′ is the Shannon–Weaver diversity index and *p_i_* is the proportion of species *i* compared to the total number of species (*S*) in the study area (or species richness of the area), which is calculated as follows: *p* (*i*) = *n_i_*/*N*
where *n_i_* is the number of individuals of species *i* and *N* is the total population (*n* individuals of all species). The Pooled InfRate Version 4.0 software was used to calculate and compare viral infection rates (IRs), or more precisely to perform viral RNA detection from tick pools [20]. The infection rates were calculated taking into account the number of pools tested, the number of ticks in each pool and the number of positive pools. Infection rates were reported per 1000 ticks for ease of interpretation. A pair of IRs was considered not to be significantly different (*p* ≥ 0.05) if the confidence interval of the difference included zero. The correlations between tick attachment sites and each of the three hosts were assessed using principal component analysis (PCA). The R software [21] (version 4.1.3) was used for statistical analyses and graphical presentations of the data.

## 3. Results

### 3.1. Tick Infestation Prevalence in Vertebrate Hosts

A total of 90 animals (30 cattle, 30 sheep and 30 goats) were examined at each site (Table 2). The total IP was 53 ± 0.03% (383/720). The tick infestation prevalence was significantly higher in cattle (IP = 92 ± 0.03%; 220/240) than in sheep (IP = 55 ± 0.1%; 131/240) and goats (IP = 13 ± 0.05%; 32/240) (*p* ≤ 0.05). A comparison between the latter two hosts showed that sheep were significantly more infested than goats were (X^2^ = 89.217, df = 1, *p* ≤ 0.05). 

In the localities of Tatki (IP = 70%), Haere Lao (IP = 61%), Saint-Louis (IP = 60%), Tessekere (IP = 52%) and Bandia (IP = 51%), more than half of the inspected animals were infested with ticks. In the other localities, the IPs were below 50% (48% in both Kassack and Kolda and 36% in Nabadji). In all the localities visited, the cattle were more affected than the sheep and goats (*p* ≤ 0.05), except in Tatki and Kassack where the IPs were comparable between cattle and sheep (*p* ≤ 0.05). Goats were the least infested hosts in these different localities, except in Nabadji and Saint-Louis where they were as infested as sheep were.

### 3.2. Specific Richness and Diversity

Table 3 shows the specific richness and diversity of the tick population collected from the different localities. The specific richness was higher in Kolda (10 species), followed by Kassack and Saint-Louis (8 species each), Bandia and Tessekere (7 species each), Haere Lao (6 species), Nabadji (5 species), and Tatki (4 species). The Shannon index indicates that the site of Saint-Louis had a higher species diversity (H = 1.68).

### 3.3. Relative Abundance of Tick Species by Host and Locality

A total of 5216, 848 and 71 ticks were collected from cattle, sheep and goats, respectively. This population included 4317 males (70.37%), 1792 females (29.2%) and 26 nymphs (0.6%) belonging to four genera (*Hyalomma*, *Amblyomma*, *Rhipicephalus* and *Boophilus*) and eleven species collected from the 720 animals that were examined (Figure 2). 

The genus *Hyalomma* was the most abundant (53.95% of the sample) and was present in all surveyed areas. Within this genus, three species were recorded, *Hy. impeltatum, Hy. marginatum rufipes* and *Hy. Truncatum*, representing 57.91%, 29.33% and 12.74%, respectively. *Hy. impeltatum* and *Hy. marginatum rufipes* were present in all study areas, while *Hy. truncatum* was absent in Kassack and Tatki. *Hy. truncatum* was collected only from cattle, while *Hy. impeltatum* and *Hy. marginatum rufipes* were present on all vertebrate hosts inspected, although mainly on cattle (Figure 2). The highest number of *Hy. impeltatum* was recorded in Tatki, while *Hy. marginatum rufipes* was mainly collected in Bandia.

The genus *Amblyomma* was represented by a single species (*A. variegatum*) and was the second most abundant genus (36.79% of the sample). This species was present at five collection sites and was collected from all three hosts, but mainly from cattle. The highest abundance of this species was recorded in Kolda (81.64%) and Bandia (64.05%).

Five species of the genus *Rhipicephalus* were collected, representing 8.51% of the tick sample. This genus was present at all collection sites. The different species recorded were *Rh. evertsi evertsi* (46.93% of this genus), *Rh. muhsamae* (29.31%), *Rh. guilhoni* (10.54%), *Rh. sulcatus* (8.43%) and *Rh. lunulatus* (4.79%). The latter was found only in Kolda and only in cattle.

The genus *Boophilus* was the least abundant, representing only 0.75% of the total sample. Two species were recorded in this genus during the study: *B. decoloratus* (91.3% of this genus) and *B. geigyi* (8.7%). The latter was present only in Kolda and was collected from cattle, while *B. decoloratus* was present in two areas (Kassack and Bandia) and was collected from all hosts, but mainly from cattle.

### 3.4. Tick Attachment Sites on Hosts

Figure 3 shows the tick species collected according to their hosts and their preferred attachment sites. The anogenital region was the most infested attachment site regardless of the host. The smaller the angle formed by two attachment sites, the higher the probability of finding the same species on these sites. In fact, the species found in the anogenital region were also found on the abdomen and feet in cattle, whereas in sheep the species found in the anogenital region were found on the abdomen, back and tail. This correlation was not observed in goats due to the small number of tick species collected from these hosts. The preferred secondary sites of tick attachment were the abdomen, tail and feet in cattle, ears, abdomen and neck in goats and ears and abdomen in sheep.

### 3.5. Detection of CCHFV in Ticks 

Of the 1956 pools tested, CCHFV was detected in 33 pools of the *Hyalomma* genus, 14 pools of the *Amblyomma* genus, 6 pools of the *Rhipicephalus* genus and 1 pool of the *Boophilus* genus.

CCHFV was detected in ticks collected from cattle and sheep. All pools of ticks collected from goats were negative for CCHFV. *Hyalomma impeltatum* accounted for 44.4% of the positive pools, followed by *A. variegatum* (25.9%) and *Rh. evertsi evertsi* and *Hy. marginatum rufipes* (11.1% for each species). *Hyalomma truncatum* and *B. decoloratus* represented less than 6% of the positive pools.

The overall IR of CCHFV in ticks was 0.15/1000 infected ticks (Table 4). The IR was higher in ticks collected from sheep (0.42/1000 infected ticks) than from cattle (0.13/1000 infected ticks) (*p* ≤ 0.05). The virus was more frequently detected in *Hyalomma* tick species, but the IR was higher in *B. decoloratus* (IR = 6.46/1000 ticks), followed by *Rh. evertsi evertsi* (3.47/1000 ticks), *Hy. truncatum* (0.26/1000 ticks), *Hy. marginatum rufipes* (0.25/1000 ticks), *Hy. impeltatum* (0.2372/1000 ticks) and *A. variegatum* (0.17/1000 ticks). The infection rates in the north (Kassack, Haere Lao, Saint-Louis, Tessekere and Tatki), centre (Bandia) and south (Kolda) were 0.58/1000 ticks, 0.41/1000 ticks and 0.142/1000, respectively. These rates were significantly higher in the north than in the centre and south (*p* ≤ 0.05). In the northern zone, the locality of Kassack had a higher infection rate (1.032/1000) compared to that in Tessekere (0.635/1000), Haere Lao (0.0483/1000), Saint-Louis (0.355/1000) and Tatki (0,223/1000) (*p* ≤ 0.05). CCHFV was not detected in any of the tick samples collected in the locality of Nabadji.

Our results showed that species of different genera (*Hy. m. rufipes* vs. *Rh. evertsi evertsi* or *A. variegatum* vs. *Hy. truncatum*) or male and female ticks of the same species collected from the same host can be infected with CCHFV. 

## 4. Discussion

The main objective of this study was to investigate emerging and re-emerging tick-borne viruses circulating in Senegal. The Crimean–Congo haemorrhagic fever virus was detected in different tick species infesting cattle and sheep in seven of the eight localities visited. The prevalence of tick infestation in livestock was 53.19%. Cattle were the most tick-infested hosts (92%), which could be explained by their less hairy skin, which is easier for ticks to attach themselves to, but also by their long-distance movements compared to those of small ruminants. They are therefore more likely to be exposed to different ecological zones suitable for the development of ticks and thereby become infested. The high infestation rate of ovine hosts compared to that of caprine hosts has been described in previous studies of tick infestation in livestock in Senegal and Pakistan [22,23]. The relatively low level of parasitism in goats could be explained by an aversion towards goats, as the quality of their fur is not favourable for tick attachment [24]. 

Although the Kolda site was richer in species, Saint-Louis was more diverse, as shown by the Shannon index. This phenomenon could be explained by the presence of species (more common in the north) in the Kolda area due to transhumance. In fact, the period of tick collection coincided with the “Tabaski” festival, when the animals are transported to the south of the country to be sold. A new report of a tick species in an area could be linked to transhumance and new ecological conditions favourable to the establishment of this species. This is the case for the *Hy. impeltatum* that was present in the Kolda area where it was absent during a study carried out by Gueye et al. (1989) [25]. *Hyalomma impeltatum* is thought to be native to the northern zone of West Africa, which is its natural habitat [26].

The diversity of 11 species in 4 genera was reported in our collections. Previous studies on livestock in the same areas showed the same high diversity with four additional species (*Hy. dromadarii*, *Hy. Impressum*, *Rh. cuspitatus* and *Rh. senegalensis*) [22,27,28]. These missing species in our study could have been related to the shorter sampling period and smaller number of sites visited compared to the longer sampling period (15 months) and the greater number of sites visited during the previous study, which allowed for a greater chance of collecting more species. Climate-related ecological changes (high temperature associated with low humidity and sparse rainfall) may also contribute to species decline [29]. 

In agreement with previous observations [22,27,28], our results confirm the absence of *B. decoloratus* in the south and *B. geigyi* and *Rh. lunulatus* in the north and centre of the country. The absence of *B. decoloratus* in the south of the country shows that its range coincides with the Sahelian xerophilic steppe and the northern Sudanian savannah in West Africa [28]. This species is replaced by *B. geigyi* in the South Sudanian and Guinean savannah of West Africa.

Cattle were infested by all tick species, mainly by species of the genus *Hyalomma* and *Amblyomma*. This trophic preference for this host may be due to its availability, the adaptation of these tick species through their morphological characteristics (the length of the hypostome) and the phenomenon of aggregation [29].

Sheep were infested by all tick species except *Hy. truncatum* and *B. geigyi. Hyalomma impeltatum* was the most abundant species collected from sheep, followed by *Rh. evertsi evertsi*. Thus, sheep seem to be the secondary host of *Hy. impeltatum* after cattle. However, sheep may be the primary host of *Rh. evertsi evertsi* given the higher infestation rate compared to that in cattle and goats.

As with sheep, all tick species collected from cattle were found in goats, with the exception of *Hy. m. rufipes, Hy. truncatum* and *B. geigyi*. Goats were mainly infested with *Rh. evertsi evertsi*. The low infestation rate by other tick species is probably due to the restricted movements of goats in the domestic environment [27]. Indeed, the presence of this species on small ruminants could indicate a domestic cycle for *Rh. everstsi evertsi*. These results are in agreement with the data obtained by Gueye et al. (1987) [28] in the Niayes area. In fact, their results showed a high infestation rate of *Rh. evertsi evertsi* in calves in permanent housing, whereas in the same study, only one *Rh. evertsi evertsi* tick was collected from native cattle in the field.

The ano-genital region was the most infested attachment site regardless of host species. Secondary preferred tick sites were the abdomen, tail and feet in cattle, the abdomen, neck and ears in goats, and the abdomen and ears in sheep. Most of the ticks collected were found on sites with little fur and thinner skin. The high tick infestation in these areas may be due to ticks’ preference for warm, moist, hidden, well-vascularised and thin sites [30,31].

Of the 11 tick species identified in this study, 6 (*Hy. impeltatum*, *Hy. marginatum rufipes*, *Hy. truncatum*, *B. decoloratus*, *A. variegatum* and *Rh. evertsi evertsi*) were naturally infected with CCHFV. These species have been found to be already naturally infected with the virus in Senegal. According to Camicas and colleagues [32], four species of these ticks can be considered major vectors of CCHFV in Senegal, namely *Hy. m rufipes*, *Hy. truncatum*, *Rh. evertsi evertsi* and *A. variegatum*. In addition, the role of vectors in the maintenance and transmission cycles of CCHFV in Senegal has been studied experimentally [33,34]. This study showed infection rates of 100% in *Hy. marginatum rufipes* and *Hy. truncatum* (one of the vectors of CCHFV in humans), 60% in *A. variegatum* and 100% for *Rh. evertsi evertsi* females after an incubation period of 15 days post-inoculation [34]. Given its high anthropophilic rate, *A. variegatum* may play an important role in human infections, often resulting in mild symptoms in areas where the tick is present [2]. Although the *Rh. evertsi evertsi* species is an inefficient vector from an epidemiological point of view, its role during epizootic periods should not be underestimated [34].

The CCHFV infection rates were higher in *B. decoloratus* than those in other species. However, its low density reflects its low involvement in the ecology of CCHFV [32].

In this study, CCHFV was most frequently detected in tick species of the genus *Hyalomma. Hyalomma impeltatum* had a relatively low infection rate compared to other tick species of the same genus. This species, although found to be naturally associated with the virus in other studies in Senegal, was removed from the list of major CCHFV vectors by Camicas and colleagues [32] because of its low epidemiological importance. However, an experimental study suggested its possible role as a vector after the detection of high levels (63%) of the CCHF viral antigen in adult ticks emerging from virus-exposed larvae [35]. Therefore, this species should be closely monitored to determine its potential epidemiological role.

The risk of CCHFV infection was higher in ticks collected from northern localities than in Central and Southern Senegal. This confirms that the northern localities of Senegal have the highest exposure to CCHFV. These results are in agreement with those of previous studies carried out in Senegal [12,13,36]. This high infection rate could be explained by the high relative abundance of the main CCHFV vectors (*Hy. m. rufipes* and *Hy. truncatum*) in the north, but also by the geographical proximity of this area to Mauritania, where the virus often circulates in a quasi-endemic form [37].

Our results showed that infection rates were higher in sheep than in cattle, which were more heavily tick-infested. These results are similar to the data obtained in Pakistan by the authors of [38,39]. Domestic ruminants, especially sheep, may play an important role in virus amplification as they become viremic in about one week [6], allowing the virus to infect more ticks during this period.

Our study showed that different species of ticks, as well as the same species of ticks of different sexes collected from a single animal host, were infected with CCHFV. These infections may be due to different modes of transmission of CCHFV, including co-feeding, transovarial and trans-stadial transmission [32,40,41]. Once infected, ticks remain positive for CCHFV throughout their lives [42]. Therefore, parallel blood sampling of livestock for the detection of CCHFV would be necessary to better understand the modes of virus transmission.

This study confirms the active circulation of CCHFV in ticks collected from livestock in different locations in Senegal, highlighting their role in the maintenance of CCHFV. Effective measures, including the development of a tick vaccine to control tick infestation in livestock, are urgently needed to prevent future CCHFV infections in humans.

## Figures and Tables

**Figure 1 tropicalmed-08-00317-f001:**
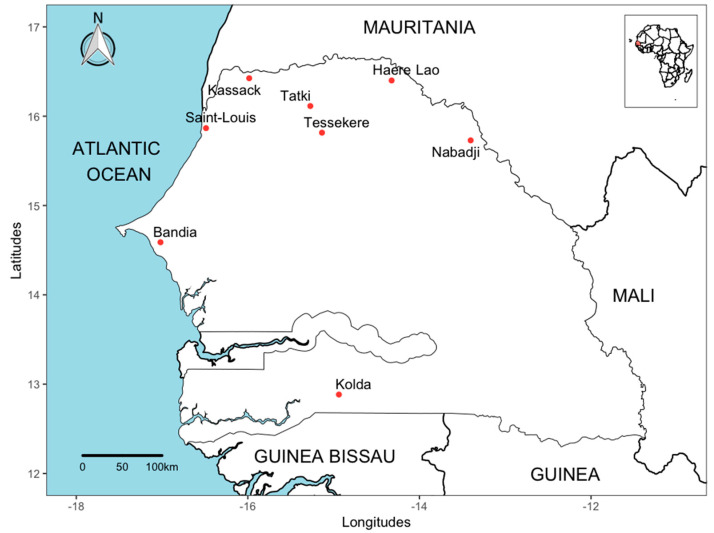
Map showing tick sampling sites in Senegal, July 2021.

**Figure 2 tropicalmed-08-00317-f002:**
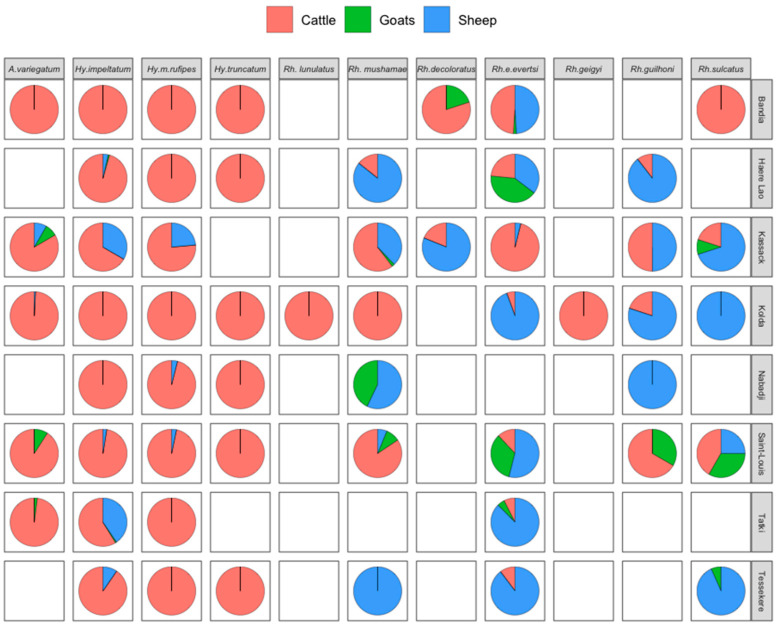
Relative abundance of tick species collected according to host and collection site, July 2021, Senegal.

**Figure 3 tropicalmed-08-00317-f003:**
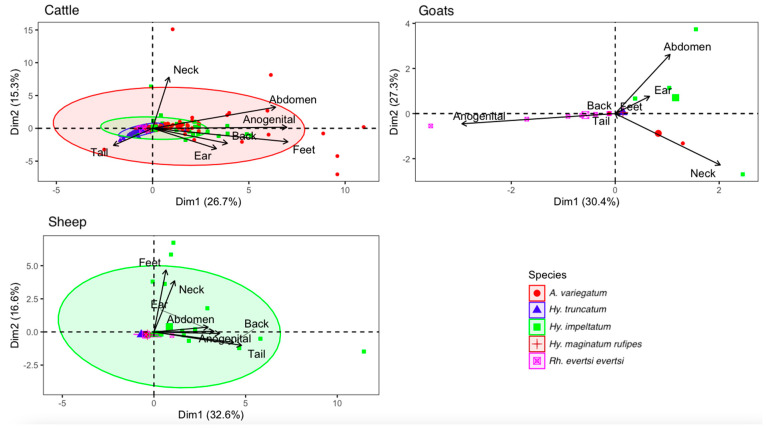
Principal component analysis (PCA) showing tick attachment sites according to species and hosts, July 2021, Senegal.

**Table 1 tropicalmed-08-00317-t001:** Geographic coordinates of the different collection sites, Senegal, July 2021.

Localities	Latitude N	Longitude W
Nabadji	15°43′44.78″	13°24′09.76″
Haere Lao	16°23′59.71″	14°19′26.91″
Tatki	16°06′49.36″	15°16′19.28″
Tessekere	15°48′55.50″	15°08′09.88″
Kassack	16°25′29.52″	15°59′03.81″
Saint-Louis	15°51′58.27″	16°29′15.26″
Bandia	14°35′19.13″	17°01′06.43″
Kolda	12°53′02.54″	14°56′19.88″

**Table 2 tropicalmed-08-00317-t002:** Tick infestation prevalence in livestock in different localities of Senegal, July 2021.

Localities	Host									Total		
	Cattle			Goats			Sheep					
	P	I	IP (%)	P	I	IP (%)	P	I	IP (%)	P	I	IP (%)
Nabadji	30	23	77	30	4	13	30	5	17	90	32	35
Haere Lao	30	30	100	30	4	13	30	21	70	90	55	61
Tatki	30	28	93	30	5	17	30	30	100	90	63	70
Tessekere	30	28	93	30	0	0	30	19	63	90	47	52
Kassack	30	22	73	30	4	13	30	17	57	90	43	48
Saint-louis	30	29	97	30	13	43	30	12	40	90	54	60
Bandia	30	30	100	30	2	7	30	14	47	90	46	51
Kolda	30	30	100	30	0	0	30	13	43	90	43	48
Total	240	220	92	240	32	13	240	131	55	720	383	53

P: prospected; I: infected; IP: tick infestation prevalence.

**Table 3 tropicalmed-08-00317-t003:** Species richness and Shannon diversity index of tick populations in the different localities of Senegal, July 2021.

Locality	Female	Male	Nymph	Abundance Total	Relative Abundance (RA%)	Shannon (H′)	Specific Richness (S)
Bandia	411	1219	3	1633	26.6	1.05	07
Haere Lao	149	336	0	485	7.91	1.24	06
Kassack	74	150	9	233	3.80	1.57	08
Kolda	373	1003	13	1389	22.6	0.74	10
Nabadji	108	183	0	291	4.74	0.68	05
Saint-Louis	152	384	0	536	8.74	1.68	08
Tatki	460	890	1	1351	22.0	0.49	04
Tessekere	65	152	0	217	3.54	1.37	07

**Table 4 tropicalmed-08-00317-t004:** Crimean–Congo haemorrhagic fever virus infection rate by tick species, location and host, July 2021, Senegal.

Localities	Species	Host	IR	95% CI	TOTAL (95% CI)
Bandia	*A. variegatum*	Cattle	0.09	(0.05–0.17)	0.14 (0.09–0.22)
	*Hy.m rufipes*	Cattle	0.18	(0.03–0.58)	
	*Hy. truncatum*	Cattle	0.25	(0.01–1.21)	
	*Rh. evertsi evertsi*	Cattle	16.3	(3.04–52.91)	
	*Rh. evertsi evertsi*	sheep	7.46	(1.36–24.41)	
Kolda	*A. variegatum*	Cattle	0.03	(0.01–0.08)	0.04 (0.01–0.09)
	*Hy. truncatum*	Cattle	0.21	(0.01–0.99)	
Haere Lao	*Hy. impeltatum*	Cattle	0.36	(0.09–0.98)	0.48 (0.19–1.00)
	*Hy. impeltatum*	Sheep	107	(8.20–422.82)	
	*Hy.m rufipes*	Cattle	0.49	(0.09–1.62)	
Kassack	*B. decoloratus*	Cattle	35.2	(2.19–158.44)	1.03 (0.19–3.33)
	*Hy. impeltatum*	Sheep	1000	(206.55–1000)	
Saint-Louis	*Hy. impeltatum*	Cattle	0.50	(0.09–1.65)	0.35 (0.02–1.94)
	*Hy.m rufipes*	Sheep	1000	(206.55–1000)	
	*Hy. truncatum*	Cattle	0.39	(0.02–1.94)	
Tatki	*Hy. impeltatum*	Cattle	0.22	(0.12–0.37)	0.22 (0.14–0.34)
	*Hy. impeltatum*	Sheep	1.16	(0.05–0.39)	
	*Rh. evertsi evertsi*	Sheep	6.17	(1.13–20.17)	
Tessekere	*Hy. impeltatum*	Cattle	0.42	(0.02–2.03)	0.63 (0.11–2.08)
	*Hy.m rufipes*	Cattle	1.49	(0.09–7.28)	
Total species	*B. decoloratus*		6.46	(0.38–30.72)	0.15 (0.11–0.19)
	*A. variegatum*		0.06	(0.03–0.10)	
	*Hy. impeltatum*		0.24	(0.16–0.35)	
	*Hy. m rufipes*		0.25	(0.10–0.53)	
	*Hy. truncatum*		0.26	0.07–0.71)	
	*Rh. evertsi evertsi*		3.47	(1.44–7.15)	
Total host	Cattle		0.13	(0.09–0.17)	
	Sheep		0.42	(0.22–0.72)	

IR: infection rate; 95% CI: 95% confidence interval.

## Data Availability

All data included in this work are provided in tables and figures.

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
