# Peer review of "Detection of Crimean–Congo Haemorrhagic Fever Virus from Livestock Ticks in Northern, Central and Southern Senegal in 2021"

_tropicalmed, 2023, doi:10.3390/tropicalmed8060317_

Round 1

Reviewer 1 Report

The discussion need deep revision and should follow a clear plan.

There are redundancy of results, please summarize and retain the most relevant for readers. 

Provide more references to discuss thoroughly the results

The manuscript need more editing and all the comments are shown on the pdf version.

The French language dominated the formulation of the text. English editing is required to improve the reading and to fit to anglophone community.

Reviewer 2 Report

Comments are suggested in the attached file.

Author Response

Thanks for the suggestions and comments. The parts underlined in yellow have been reviewed and corrected. See the revised manuscript.

Reviewer 3 Report

Self-citation should be reduced

Should be read by a native speaker for improvement of the English language. 

Author Response

Reviewer #3 : Minor editing of English language required

We thank the Reviewer for the positive comments and suggestions. We have addressed his/her concerns in the revised manuscript.

  English language was edited. Se the revised manuscript.

Reviewer 4 Report

The manuscript by BADJI et al reports the analysis of ticks collected in several area of Senegal  to determine the diversity of tick species, their infestation rates in livestock and their infection by CCHFV.

The manuscript is well written e provide further data on the presence of CCHFV in Senegal but without any significant novelty in comparison to literature data. To increase the relevance of the study the genotyping of CCHFV should be added.

Minor:

Write the name of each tick genus in the abstract should be write in italics (lines 19-20)

Line 95: CRORA, 2021. Add the reference

Line 240, please delete “while all ticks collected from caprine were negative for CCHFV”. This info was also reported few lines above.

Lines 289-290. Please, check the sentence and rewrite to increase readability

Line 295: write impeltatum in italics

Line 340: references have to be indicated as numbers.

Line 355: delete the word “antigens”, test could be performed to detect antigens or viral RNA

None

Author Response

Reviewer #4:

The manuscript by BADJI et al reports the analysis of ticks collected in several area of

Senegal to determine the diversity of tick species, their infestation rates in livestock and their

infection by CCHFV.

The manuscript is well written and provide further data on the presence of CCHFV in Senegal

but without any significant novelty in comparison to literature data. To increase the relevance of the study the genotyping of CCHFV should be added.

Response: Dear reviewer thank you for your comments. Due to the short delay of the viral genotyping cannot be added in this revised manuscript. However, sequencing and genetic analysis of these strains are ongoing and will be published in another manuscript.

Write the name of each tick genus in the abstract should be write in italics (lines 19-20) 

Response: We corrected. Se the revised manuscript L19-20.

Line 240, please delete “while all ticks collected from caprine were negative for CCHFV”.

Response: We corrected. Se the revised manuscript L247.

Lines 289-290. Please, check the sentence and rewrite to increase readability

Response: We corrected. Se the revised manuscript L302-303.

Line 295: write impeltatum in italics

Response: We corrected. Se the revised manuscript L308.

Line 340: references have to be indicated as numbers.

Response: We corrected. Se the revised manuscript L355.

Line 355: delete the word “antigens”, test could be performed to detect antigens or viral RNA

Response: We deleted. Se the revised manuscript L366.

Round 2

Reviewer 4 Report

Authors modified the manuscript as requested. It is a pity do not have genetic data to track CCHFV diffusion in the different regions of Senegal to better clarify the route of CCHFV diffusion providing a complete picture.